# Deeply in Plasticenta: Presence of Microplastics in the Intracellular Compartment of Human Placentas

**DOI:** 10.3390/ijerph191811593

**Published:** 2022-09-14

**Authors:** Antonio Ragusa, Maria Matta, Loredana Cristiano, Roberto Matassa, Ezio Battaglione, Alessandro Svelato, Caterina De Luca, Sara D’Avino, Alessandra Gulotta, Mauro Ciro Antonio Rongioletti, Piera Catalano, Criselda Santacroce, Valentina Notarstefano, Oliana Carnevali, Elisabetta Giorgini, Enrico Vizza, Giuseppe Familiari, Stefania Annarita Nottola

**Affiliations:** 1Department of Obstetrics and Gynecology, Università Campus Bio Medico di Roma, Via Álvaro del Portillo, 21, 00128 Rome, Italy; 2Department of Clinico-Surgical, Diagnostic and Pediatric Sciences, Faculty of Medicine and Surgery, University of Pavia, Via Alessandro Brambilla, 74, 27100 Pavia, Italy; 3Department of Life, Health and Environmental Sciences, University of L’Aquila, Via Vetoio, Loc. Coppito, 67010 Coppito, Italy; 4Department of Anatomy, Histology, Forensic Medicine and Orthopaedics, Sapienza University, Via A. Borelli, 50, 00161 Rome, Italy; 5Department of Gynecology and Obstetrics of “San Giovanni Calibita” Fatebenefratelli Hospital, Isola Tiberina of Rome, Via di Ponte Quattro Capi, 39, 00186 Rome, Italy; 6Department of Pathological Anatomy of “San Giovanni Calibita” Fatebenefratelli Hospital, Isola Tiberina of Rome, Via di Ponte Quattro Capi, 39, 00186 Rome, Italy; 7Department of Life and Environmental Sciences, Università Politecnica delle Marche, Polo Montedago Via Brecce Bianche, 60131 Ancona, Italy; 8Gynecologic Oncology Unit, Department of Experimental Clinical Oncology, IRCCS Regina Elena National Cancer Institute, Via Elio Chianesi, 53, 00144 Rome, Italy

**Keywords:** microplastics, human placenta, electron microscopy, cell organelles

## Abstract

**Highlights:**

**What are the main findings?**
For the first time, microplastics were detected and localized by electron microscopy in human placentas.The presence of microplastics was correlated with ultrastructural alterations of some cell organelles in placental tissue, mainly in the syncytiotrophoblast.
**What is the implication of the main finding?**
Microplastics in human placentas could contribute to the activation of pathological traits, such as oxidat ive stress, apoptosis, and inflammation.Microplastics in human placentas may cause long-term effects on human health.

**Abstract:**

Microplastics (MPs) are defined as plastic particles smaller than 5 mm. They have been found almost everywhere they have been searched for and recent discoveries have also demonstrated their presence in human placenta, blood, meconium, and breastmilk, but their location and toxicity to humans have not been reported to date. The aim of this study was twofold: 1. To locate MPs within the intra/extracellular compartment in human placenta. 2. To understand whether their presence and location are associated with possible structural changes of cell organelles. Using variable pressure scanning electron microscopy and transmission electron microscopy, MPs have been localized in ten human placentas. In this study, we demonstrated for the first time the presence and localization in the cellular compartment of fragments compatible with MPs in the human placenta and we hypothesized a possible correlation between their presence and important ultrastructural alterations of some intracytoplasmic organelles (mitochondria and endoplasmic reticulum). These alterations have never been reported in normal healthy term pregnancies until today. They could be the result of a prolonged attempt to remove and destroy the plastic particles inside the placental tissue. The presence of virtually indestructible particles in term human placenta could contribute to the activation of pathological traits, such as oxidative stress, apoptosis, and inflammation, characteristic of metabolic disorders underlying obesity, diabetes, and metabolic syndrome and partially accounting for the recent epidemic of non-communicable diseases.

## 1. Introduction

Over the past century, global plastics production has grown exponentially to over 350 million tons per year produced worldwide, part of which ends up polluting the environment [1]. Microplastics (MPs) are defined, according to European Food Safety Authority (EFSA), as plastic particles smaller than 5 mm [2,3]. They are produced at this size or result from the fragmentation of larger plastic structures. MPs are found in aqueous, terrestrial [4,5], and airborne [6] environments. In addition, there have been several reports of MPs in food, particularly seafood [7,8], sea salt [9,10], and drinking water [11,12,13]. MPs are primarily detected in the gastrointestinal tract of marine animals [14], while the cellular uptake and accumulation of MPs in tissues have been demonstrated in experimental settings [15,16,17]. Scientists and public authorities have raised concerns about MPs in food, potential human intake, and health consequences [2,18,19], but data are scarce. MPs have also recently been found in human tissues, fluids, and secretions [20,21,22,23], but the effect of MPs on a global and cellular level is still unclear.

Microplastic pollution in the oceans affects the marine ecosystem, coastal tourism, and even human health. Animals living in the oceans, such as fish, whales, seabirds, and turtles, unknowingly ingest plastic [24]. Many marine organisms swallow pieces of plastic, and most ingested MPs accumulated especially in gastrointestinal tracts of biota. The number of MPs in biota is very variable, oscillating between one and 20 items in different types of fishes, also according to different types and forms of the MPs [25].

MPs can be found in either the abiotic or biotic compartments, but the biosphere must be understood as a whole. The biogeochemical cycles of plastics are particularly complex [26]. In humans, MPs enter the organism through three routes: gastrointestinal (food), respiratory, and cutaneous [27]. Through the blood circulation, they are deposited mainly in organic tissues [28]. Depending on their dimensions and shape, it is hypothesized that they can spread through passive diffusion or active phagocytosis [29]. Within organic tissues, MPs are considered foreign bodies by the host organism and as such trigger local immunoreactions. MPs can be a carrier for other chemicals, such as environmental pollutants and plastic additives, whose harmful effects are well known, inducing the so-called “trojan horse effect” [2,15,17,30,31,32]. MPs and plastic additives often act as endocrine disruptors, i.e., exogenous substances or mixture of substances that alter the functions of the endocrine system and harm the health of the entire organism, its progeny, or certain cell populations of the organism itself [33].

What is known about the effects of MPs on pregnant health and fetal development is largely derived from studies on animal models. In animal models, various evidence of toxicity has been reported. MPs were found to interfere with energy production and lipid metabolism, increasing oxidative stress and neurotoxic response [34]. Their presence has been associated with toxic effects on cell cultures, causing apoptosis, inflammation, mitochondrial and lysosomal dysfunction, and genotoxicity [35]. Their interaction with the immune system appears to be disruptive, with alterations on a genetic level for the expression of genes involved in the immune response.

MPs can cause changes on a phenotypic level in mice, as well as in the expression of genes and epigenetics, as was demonstrated by brain abnormalities in mouse pups whose mothers were fed with plastic microparticles [36]. Cognitive capacity alterations were discovered, associated with a modification of RNA expression, and immunofluorescence revealed MP infiltrates in the brain of pups. Jeong et al. (2022) [36] showed that the maternal administration of polystyrene nanoplastics (i.e., plastic fragments with size < 100 nm), during gestation and lactating periods altered the functioning of neural cell compositions and brain histology in progeny. Polystyrene (Ps) nanoplastics also induced molecular and functional defects in cultured neural cells in vitro. The abnormal brain development caused by exposure to high concentrations of polystyrene nanoplastics results in neurophysiological and cognitive deficits in a gender-specific manner in mice. Adult offspring of female mice exhibit short- and long-term deficient social recognition, reduced sociability, and increased repetitive behavior when they were exposed to the Polybrominated diphenyl ethers. Exposure to Polybrominated diphenyl ethers during intrauterine development produces neurochemical, olfactory, and behavioral processes that are relevant and very similar to those of autism spectrum disorders (ASD) in humans. These effects can reprogram early neurological development within central memory and social networks [37]. Oral administration of monodispersed polystyrene also causes damage to the visceral organs in mice. The main toxic effects are the damage to the liver function and the lipid metabolism abnormality. Chronic exposure to monodispersed polystyrene significantly increases plasma glucose levels and ROS levels but does not influence plasma insulin secretion. Ultimately, oral administration of monodispersed polystyrene increases ROS, liver triglycerides and determines the accumulation of cholesterol in mice [38].

MPs have been found in human placenta and in the meconium of newborns [20,23] even in higher concentrations than in adults’ stool [39], in the breastmilk [22], and in the blood [21], confirming that the exposure to MPs begins indeed in the earliest stages of human life.

If MPs interact with human placental cells, being capable of altering energy pathways as they do in animal models, there could be numerous concerning consequences.

The aim of this study was twofold:To locate MPs within the intra/extracellular compartment in human placenta by variable pressure scanning electron microscopy (VP-SEM) and transmission electron microscopy (TEM).To understand whether their presence and location are associated with possible structural changes of cell organelles.

## 2. Materials and Methods

Our scientific hypothesis is that MPs, ubiquitarian in the environment, can also be found in different placental cell domains, inducing organelle alterations and the consequent appearance of pathological traits in the newborn. For this reason, ten women who delivered at the “San Giovanni Calibita” Fatebenefratelli Hospital were selected, from July 2021 to September 2021, to participate in the study after providing appropriate informed consent. The study was approved by ethics Committee Lazio1 (Prot. N. 352/CE Lazio1 of 31 March 2020).

### 2.1. Experimental Design, Enrolment of Patients and Placentas Collection

To represent the various modalities of childbirth, placentas were sampled both from cesarean sections and from vaginal births, in women with single fetus and low risks pregnancies.

The exclusion criteria adopted were:-Particular diets prescribed by a physician in the four weeks prior childbirth.-Diarrhea or constipation in the two weeks prior to childbirth.-Antibiotic use in the two weeks prior childbirth.-Drug use interfering with intestinal absorption (e.g., active carbon or cholestyramine) in the two weeks prior to childbirth.-Diagnosis of a gastrointestinal disease (e.g., ulcerative colitis or Crohn disease, cancer, organ transplant, HIV or any other disease that requires medical treatment).-Invasive or abrasive odontoiatric treatments in the two weeks prior to childbirth.-Current or recent participation (within the four weeks prior childbirth) to a clinical experiment.-Alcohol abuse (defined by a test score for identification of alcohol abuse disorders > 10).

Each woman was given a questionnaire that aimed to monitor their possible plastic consumption, based on products used for cosmetics, personal hygiene, and alimentation. The questionnaire also collected information regarding age, smoking habits, alcohol consumption, and any pathophysiological condition (diabetes mellitus, hypertension, etc.) already existing or acquired during pregnancy (Table 1).

After childbirth, the placentas were collected using a plastic-free procedure, sampled, and stored for their observation and analysis under scanning and transmission electron microscopes. To avoid contamination with plastic or synthetic fibers, a plastic free-kit was used, and the sampling was conducted according to the specific plastic-free protocol fully reported in our previous paper [20] and briefly described below.

For the cesarean section delivery, aseptic techniques used were the customary ones. Cotton sheets were used for the sterile operatory field and the operators used cotton gloves during the surgical procedure. After birth, the umbilical cord was tied off with a metal clamp. The operators proceeded carefully with the collection of the placenta after expulsion, without ever introducing their hands to the uterus.

For vaginal birth, assistance techniques were the usual ones, but the midwife used cotton gloves during the final phases of expulsion of the fetus. Cotton sheets were used under the mother, and the graduated bag for blood collection was used only after afterbirth and the collection of the placenta. After delivery, the umbilical cord was tied off with a metal clamp. Once severed, the umbilical cord was put on the cotton sheet, carefully avoiding any contact with plastic materials.

The placenta was then stored in a metallic container to be transported in ice to the Pathological Anatomy Department within 10 min from delivery.

The pathologist wore cotton gloves and used a metallic scalpel to proceed in cutting the section from the maternal side, carefully collecting the sample exclusively from the intraparenchymal portion. From each placenta, three samples were collected, from the same portion, for a total of 30 samples, with sizes of 5 mm for electron microscopy studies. The samples, catalogued as A, B, and C, were subsequently stored in glass containers, without plastic gaskets or any other trace of plastics, at a temperature between 0 °C and 7 °C.

### 2.2. Electron Microscopy Analysis

All samples were fixed with 2.5% glutaraldehyde (Electron Microscopy Sciences, Hatfield, PA, USA) in phosphate buffered saline solution (PBS) for at least 48 h at 4 °C. Plastic-free procedures for sample preparations were adopted to avoid contamination. Routine plastic tools were substituted with metallic or glass ones.

#### 2.2.1. Variable Pressure Scanning Electron Microscope

A variable pressure scanning electron microscope with dual energy dispersive X-ray spectroscopy detectors (VP-SEM-dEDS, Hitachi SU-3500, Tokyo, Japan) set in parallel disposition (Bruker, XFlash^®^660, Billerica, MA, USA) was used to investigate the placental villi morphology and the presence of non-organic microparticles. Observations were carried out on at least five millimetric fragments from each sample, with regard to the external surface of the villi. The samples destined for morphological observation of villous structures underwent gentle washing with saline solution NaCl 0.9%, whereas the samples destined for analysis for MPs detection were left unwashed.

Hydrated samples devoid of conductive coating were deposited on aluminium stub to be observed and analysed. To minimize water loss during low vacuum operations, a thin coat of distilled water was poured onto the hydrated samples at room temperature. The use of VP-SEM-dEDS in combination with Peltier cooling stage control permitted imaging of hydrated specimens under variable pressure conditions. The surface collapse was prevented by carefully regulating the chamber pressure and cooling temperature. To avoid radiation damage and drifting images, all samples were examined at an accelerating voltage based on the pressure/temperature parameters, as reported elsewhere in our previous studies [40].

In comparison to light microscopy, SEM is capable to provide well-resolved direct imaging of the particles lying on the villous surface of the placenta. SEM microscopes are equipped with of ultra-variable-pressure (UVD) and backscattered compositional (BC) detectors for imaging observation. The EDS detectors allow to acquire X-ray of different elements into an energy spectrum for quantitative chemical compositional and/or multi-mapping imaging analyses. Due to the simultaneous collection of images and elemental mapping, the use of VP-SEM-dEDS aids in further distinguishing bio-organic materials from possible MPs contaminations.

#### 2.2.2. Transmission Electron Microscopy

Fixed samples destined for TEM observations were reduced in smaller fragments (size 1–2 mm). After several washings in PBS, they were post-fixed with a 2% osmium tetroxide (Electron Microscopy Sciences) in PBS for 2 h, in a dark compartment at 4 °C and incubated with 1% tannic acid (Sigma-Aldrich, St. Louis, MO, USA) in distilled water for 40 min, also in a dark compartment at room temperature (RT). The samples were then rinsed in PBS, dehydrated through ascending series of ethanol, immersed firstly in propylene oxide (Sigma-Aldrich) for 40 min, and then left overnight in a propylene oxide/resin 1:1 solution. Finally, they were embedded in Agar 100 resin (Agar Scientific, Stansted, UK) for 48 h at 60 °C.

Resin blocks were sectioned using an Ultracut E ultramicrotome (Leica EMUC6, Wetzlar, Germany). For light microscopy (Zeiss Axioscope, Gottingen, Germany) observation, methylene blue staining (Sigma-Aldrich) was used for coloration of semithin sections (1 μm); for TEM (Zeiss EM10, Oberkochen, Germany), operating at 60 kV, ultrathin sections (90–100 nm) were cut with a diamond knife, mounted on copper grids, contrasted with Uranyless (Uranyl acetate alternative) (TAAB Laboratories Equipment Ltd., Aldermaston, UK) and lead citrate (Electron Microscopy Sciences), and photographed. Further details of sample preparation for TEM observations have been reported elsewhere, in our previous studies [41,42].

A TEM semi-quantitative analysis was performed by observing three different grids containing three placenta ultrathin sections for each sample (Table 2).

## 3. Results and Discussion

In this study, VP-SEM and TEM, allowing surface and in depth morphological ultrastructural analysis of placental tissue, led to the detection of MPs in the intra/extracellular compartment, revealing the modifications to the cellular microstructure induced by them.

### 3.1. Results: Morphological and Qualitative Analyses by VP-SEM-dEDS

Morphological and chemical characterizations of different regions of villous tree in human placenta were investigated in their hydrated states using low vacuum VP-SEM-dEDS (Figure 1), to reveal directly by imaging and tracking techniques possible elusive MPs contaminations in different micrometric regions. Figure 1a clearly shows a stem villus with mature intermediate and terminal villi. By magnification of a particular region of the stem villus trunk, different dark spots, indicated by blue arrows, are visible on its surface in Figure 1b. To identify the chemical characterization of the dark spots, a further magnification of a particular region, exhibiting two dark particles of 4.8 and 3.7 µm in size, was probed following the EDS technique (Figure 1c). The morphometric dimension of the dark particles has been correlated with their chemical constituents by EDS spectra reported in Figure 1d. The chemical spatial distribution using EDS multi-elemental images was capable to identify micrometric contaminants constituted of only carbon, leaving empty areas in nitrogen and oxygen field of detection as shown in Figure 1e–g (C, N, O are referred both to the chemical element and EDS mapping images). The resulting probed micrometric region constituted of carbon, nitrogen, and oxygen belongs instead to the natural biological tissues, the placenta. The remaining peaks of the EDS spectra shown in Figure 1d are attributed to the background environment of the experimental VP-SEM technique (Na, P, Cl, and Al). Interestingly, the mapping image of Figure 1e clearly shows a densely packed atomic carbon micro-area (intense blue colour) compared to the remaining placenta area. Plastic is known to be composed of packed carbon polymers, and in this regard, this finding is compatible and can be attributed to the existence of a particular species of microplastic. 

Other inorganic micro-objects interpretable as MPs, sometimes multi-layered, varying in size from 2.1 to 18.5 µm but showing morphology (irregularity in the shapes, borders, and surfaces) and chemical composition (appearance as compact carbon micro-objects) comparable to those of the microparticles shown in Figure 1, have also been observed in other regions of the villous tree of all the samples observed. Particles smaller than 2.1 µm were probably present. Their dimension not only made them more difficult to detect, but it also required a very focused electron ray probed by the EDS technique, not which the hydrated sample cannot withstand. Thus, confirmation of their presence was not possible.

### 3.2. Results: Observations by TEM

TEM analysis of ultrathin sections of 10 samples of placenta demonstrated the presence of particles compatible with MPs inside all the different compartments of villi of human at term placenta.

Figure 2 shows the ultrastructure of a terminal villus in a human at term placenta, including: The syncytiotrophoblast, i.e., the most external layer that is richly covered by microvilli and separates the villous interior from the maternal blood; The cytotrophoblast, which lies underneath the syncytiotrophoblast and is composed of single cells that support the growth and regeneration of the latter. The cytotrophoblast is not always present at the end of gestation and sporadic cells are visible throughout the placenta. Syncytiotrophoblast and cytotrophoblast layers are separated from the stromal connective tissue by a thin basement membrane. Fetal vessels (capillaries or sinusoids) are in the innermost compartment. This layering represents the same basic structure common to all types of villi.

Figure 3 shows the presence of particles compatible with MPs in ultrathin sections of a human placenta sample photographed at TEM. MPs are observed inside the syncytiotrophoblast layer either free in the cytoplasm (a) or encapsuled in structures beneath the plasma membrane easily mistakable for organelles, such as vacuoles (b), lipid droplets (c), vesicular bodies (d), lysosomes (e), or peroxisomes (f), as morphologically described by electron microscopy. The contents of these last two organelles usually show the same electron density, so they cannot be unequivocally distinguished by morphological features from both each other and other vesicular structures in the cell, and cytochemical methods must be employed for their identification. In all samples analyzed, moderately electrodense and highly dilated vesicles of endoplasmic reticulum, swollen mitochondria, whorled membranous bodies, and autolysosomes containing mitochondria, mitochondrial remnants (b), and other cell debris also occur in association with the presence of MPs.

Figure 4 shows the presence of MPs also in the inner section of the villus (a), particularly in the pericytes (b, c) and endothelial cells (d, e) surrounding fetal microvessels, indicating that the MPs can pass through the placenta tissue and reach the fetal blood. 

Figure 5 shows ultrastructural changes in some organelles of syncytiotrophoblast cells possibly correlated to the presence of MPs. Particularly, the rough endoplasmic reticulum was not composed of narrow and parallel cisternae (orthodox shape) but appears to consist of many communicating dilated vesicles, with a secretory material, moderately electrodense, in the lumen, sparsely populated by ribosomes (a). Dilatation of endoplasmic reticulum is usually accompanied by degranulation and disaggregation of polyribosomes that are free in the cytoplasm (b). In association with the dilated endoplasmic reticulum, swollen electrodense mitochondria next to smaller, pycnotic, and electrodense ones and lipid droplets were observed (c, d).

Structural changes of the same organelles and the presence of whorled membranous bodies probably derived from involuting endoplasmic reticulum, mitochondria, autolysosomes containing mitochondria, mitochondrial remnants, and other structures that also appear in the cytotrophoblast cells, as demonstrated in Figure 6a–c. Endoplasmic reticulum dilated vesicles and swollen electrodense mitochondria also occur in endothelial cells surrounding the sinusoids (6d). 

Figure 7 highlights the presence of a MP (b, c) inside the endothelial cell in combination with the narrowing of a fetal capillary. The nucleus of the endothelial cell appears enlarged, compressing the adjacent vessel.

A detailed semi-quantitative description of the ultrastructural features representative of the general situation shown by the micrographs is summarized in Table 2.

TEM semi-quantitative analysis was performed by observing three different grids containing three placenta ultrathin sections for sample. A score (+++ = marked presence; ++ = presence; + = scarce presence; - = absence) is provided for the considered parameters. The table shows that MPs are present in all samples due to the use of food in plastic wrapping, drinks in plastic containers, cosmetics, and toothpastes with synthetic polymers, listed in Table 1. Plastics are found localized in both intracellular and extracellular compartment and have been shown to relate to endoplasmic reticulum and mitochondria damage, and to microvilli modifications as well. Only slight, not significant variations in MP amount among the ten assessed placentas are observable. For this reason, concerning MP deposition and amount in placental tissues, neither gender differences nor differences between vaginal and cesarean delivery have been noticed. In this regard, further studies carried out on a larger number of samples seem advisable.

### 3.3. Discussion

In our previous study [20], MP fragments were identified by Raman microspectroscopy in the placenta of pregnant women. For this reason, the term “plasticenta” was coined. The present study for the first time demonstrates the presence of MPs in the different compartments of human placenta at the microscopic scale. Using VP-SEM-dEDS on hydrated samples and TEM analysis on ultrathin sections of the same placental tissue, particles compatible with MPs were found on the surface of placental villi, inside cells of different placenta cellular layers, or in the extracellular environment. SEM imaging revealed carbon micro-objects of variable size and shape that are compatible, according to their chemical profiles, with possible MP fragments. These chemical profiles are in fact comparable to those shown by plastic fragments detected and described in other studies [43]. We were able to identify the possible MPs on a micrometric area of placenta performing the chemical identification with EDS technique, which provides concrete morpho-chemical results. This analysis was supported by other studies that evaluated the morphological characteristics by SEM and the EDS chemical spectrum of MPs, even if their morphology may vary depending on the environment in which they have been found [44,45,46]. In particular, we were able to discern possible carbon MP fragments from organic material or mineral crystals by chemical bidimensional mapping, since MPs have a high carbon packing density compared to the surrounding hydrated placenta with low packing density of carbon, nitrogen, and oxygen atomic species interacting with molecular water. These fragments were found in the surface of various villous typologies (stem, intermediate, terminal). We can safely exclude contamination of the samples thanks to the scrupulous methodology used for collection and sampling. The samples were taken from the innermost sections of the cotyledons, never touched, never exposed to air if not instantly prior to fixation, and never interacted with any plastic material. The presence of such fragments in the villous surface can then suggest a dispersion of possible MPs in maternal blood, through which they can reach the placenta. In fact, villous trees protrude into the lacunae of the maternal blood, easily entering into contact with MP fragments that may deposit on their surface. SEM analysis was also important because it was conducted on the same micro-area of tissue that would also undergo TEM analysis for each patient, so the EDS analysis could provide preliminary information on whether possible plastic particles were present in that segment. It has already been shown that syncytiotrophoblast is the key player in regulating nanoparticle transport across the human placenta [47]. Our results indicate that this is probably also the case for MPs. In fact, the analysis of villous cellular ultrastructure by TEM suggested that MP fragments were mostly, though not exclusively, found in the syncytiotrophoblast. MPs were also detected in the basement membrane, the cytotrophoblast, pericytes and endothelial cells surrounding the fetal capillaries. They have been found free in the cytoplasm as well as inside lipid membranes, easily confusable with organelles, such as vacuoles, lipid droplets, multivesicular bodies, lysosomes, and peroxisomes. This is the probable reason why until now nobody had noticed their presence in the intracellular compartment as Johann Wolfgang von Goethe said:” We only see what we know”. We hypothesized that MPs may be uptaken by the tissue and encapsulated inside the organelles after reaching the placenta through the gastrointestinal and/or respiratory tracts and skin [27].

In all placental samples analyzed by TEM, a dilated rough endoplasmic reticulum, partitioned into many vesicles communicating with each other, has been mainly found in the syncytiotrophoblast. These vesicles are discretely electron-dense, possibly due to the presence of secretory material in the lumen and are sparsely populated by ribosomes. Degranulation and disaggregation of polyribosomes accompany endoplasmic reticulum dilatation, resulting in free polyribosomes in the cytoplasm. Under conditions of optimal fixation, the endoplasmic reticulum should instead appear as an extensive network of membranes formed by stacks of narrow and parallel cisternae. Although a delay in fixation can quickly lead to dilation, caused by hypoxia after conventional postpartum due to a lack of placental perfusion [48], we exclude that reticulum dilation is due to a delay in fixation, as the samples were carefully handled and fixed within the appropriate timeframe (less than 10 min) after birth.

We believe that endoplasmic reticulum dilatation could be a sign of endoplasmic reticulum stress, which has a significant impact on syncytiotrophoblast function through activation of the unfolded protein response (UPR), which inhibits new protein translation. The unfolded proteins accumulate in the lumen, making the cisternae appear more electrodense. Due to the diminished protein synthesis, ribosomes are less recruited to the endoplasmic reticulum surface; within the syncytiotrophoblast’s cytoplasm of the samples examined, free ribosomes were indeed abundant. In circumstances of severe endoplasmic reticulum stress, the cisternae’s surface may have few, if any, ribosomes attached, giving the appearance of simple vesicular structures harboring a proteinaceous precipitate. Our hypothesis agrees with the observations of Wang et al. (2021) [49] about endoplasmic reticulum stress and mitochondrial dysfunction induced by the presence of MPs in human HK-2 kidneys cells and in the kidneys of mice. In support of our assumption is also the presence of aggresomes in some of the analyzed samples. Aggresomes are non-membraned pericentrosomal cytoplastic structures that harbor polyubiquitinated misfolded proteins, chaperones, and proteasomes. Their formation represents a cytoprotective mechanism in which misfolded protein complexes are pushed to juxtanuclear sites and encased in intermediate filaments; they are linked to a ubiquitin-proteasome system dysfunction [50,51,52,53]. Hollóczki (2021) [54] has demonstrated that nanoplastics can influence the relative stability of secondary structure of proteins resulting in protein misfolding.

At a phenotypical level, endoplasmic reticulum stress in the placenta is linked to the pathophysiology of human intrauterine growth restriction, preeclampsia, early pregnancy loss, placental ageing, and stillbirth in the placenta [55,56,57,58,59]. This is not the case here, as the selected pregnancies all occurred normally, and the infants were all healthy. The presence of MPs could induce endoplasmic reticulum stress in human placenta, and there could be a correlation with mitochondrial stress, given that the two often share the same etiology [60,61]. It is known that mitochondrial and endoplasmic reticulum function are linked via membrane junctions known as mitochondria-associated endoplasmic reticulum membranes (MAMs). Calcium transporters and ion channels are concentrated at these sites, and calcium flux between the two organelles is bidirectionally linked to their functionality. Loss of calcium homeostasis in the endoplasmic reticulum impairs the protein folding machinery, resulting in an accumulation of unfolded or misfolded proteins in the lumen and endoplasmic reticulum stress. Similarly, faulty mitochondria may cause an accumulation of unfolded and misfolded proteins within their matrix, resulting in stress [60].

In all examined samples, near to endoplasmic reticulum vesicles, we found mitochondrial swelling (ballooning cristae or intracristal swelling) next to mitochondrial pyknosis, several myelin figures arising from endoplasmic reticulum, including whorled membranous bodies, which are likely derived from involuting mitochondria (Figure 6a–c), and autolysosomes containing mitochondrial remnants and other structures. Interestingly, Liu et al. (2022) [44,62] and Ding et al. (2021) [63] also observed changes in mitochondrial ultrastructure and swelling of mitochondrial cristae in mouse spermatocyte line GC-2 and gastric epithelial line GES-1 treated with polystyrene MPs. Mitochondrial swelling and consequent dysfunction are notably involved in the pathogenesis of many human diseases associated with oxidative stress, such as ischemia (infarction)-reperfusion, hypoxia, inflammation, and diabetes, among others. Regulation of the mitochondrial matrix volume may provide relief to stress, which would allow mitochondria to maintain their functional and morphological integrity [64]. In some of the observed samples, mitochondrial matrices contain prominent electron-dense granules, and entire mitochondria can be packed with such granules (Figure 5a). A rise in the density, size, and number of dense granules has been reported in a variety of pathological states [65], and mitochondrial dysfunction can lead to an abnormal accumulation of these granules [66]. These granules indeed could be the result of calcium accumulation in the mitochondria. In fact, an increase in intramitochondrial of calcium concentration cause swelling [64] and when mitochondria were incubated in calcium-containing media, this ion was concentrated in dense granules, increasing their size and density [65]. We can then speculate that the observed increase in density and size of dense mitochondrial granules could be related to endoplasmic reticulum/mitochondrial stress. Mitochondrial dysfunction could cause lipid droplet formation as a generalized stress response, preventing lipid peroxidation and removing damaged proteins and lipids [67,68]. The latter is consistent with our findings of lipid droplets near endoplasmic reticulum dilated vesicles and pyknotic mitochondria. MPs can be degraded to nanoplastics, even within tissues. In a recent study Shen et al. [69] have shown that polystyrene nanoplastics of different sizes (25, 50, 100, and 500 nm) and surface charges had a specific toxicity pattern on human placental cells that depended on size and surface charge. Smaller the size of the polystyrene nanoplastics, greater is the toxicity induced on human placental cells. The induced toxicity ranged from the inhibition of the activity of the protein kinase, to the increase in oxidative stress, up to the arrest of the cell cycle. The authors noted an increase in the presence of ROS in exposed human placental cells. This is known to cause DNA damage and lead to cell cycle arrest in the G1 or G2 phase, inflammation, and apoptosis.

In this study, we consider the role that particles compatible with MPs may play in placental stress in terms of endoplasmic reticulum dilation/stressed mitochondria/oxidative stress. Numerous in vivo and in vitro studies have shown that MPs cause oxidative stress, apoptosis, inflammation, mitochondrial and lysosomal dysfunction, autophagy, and genotoxicity in various tissues (gut, intestine, liver) and cell types (human keratinocytes, human intestinal epithelial cells, gastric epithelial cells, human umbilical vein endothelial cells) [35,63,70,71,72]. Abnormal colocalization of mitochondria and endoplasmic reticulum has been also found in porcine oocytes exposed to Monobutyl phthalate (phthalates are a group of chemicals associated with PVC products as plasticizers in medical tubing, soft plastic toys, food packaging and in many personal care products) [73].

Although the insults determined by MPs can be repetitive, there is a specificity related to the cell types subjected to the insult of MPs. Gautam et al. (2022) [74] have cultured six different cell lines with different concentrations of Polyethylene MPs. Authors found that Polyethylene MPs do not have the same effects on all kinds of cells and tissues exposed, and the immune modulation is not necessarily inflammatory. The demonstration that the effects of some MPs are organ specific is very important, since placental cells, especially those of the cytotrophoblast, have their own specificities; they govern and regulate the metabolic systems that the fetus will have to use after birth to deal with the environment outside the uterus.

Alteration in mitochondrial production of ROS, metabolic intermediates, or energetic compounds, such as α-ketoglutarate, ATP, or acetyl-CoA, drives short and long-term changes in mtDNA and nDNA transcriptional activity through both signal transduction and epigenetic regulatory processes. ROS production and the redox state of the cell are directly tied to epigenetic processes [75], particularly during fetal development. What is described for the mitochondria may also be true for the dysfunctions found in the other intracellular organelles that could condition alterations in the expressiveness of DNA in the long term. This could lead to alteration in fetal developmental programming, which determines adaptation from an evolutionary perspective, conferring a tradeoff that favors short-term survival/reproductive fitness at the long-term cost of disease susceptibility, particularly for complex, common disorders [76].

Huang et al. [77] evaluated the effects of PS on insulin sensitivity in mice fed with normal chow diet (NCD) or high-fat diet (HFD). Mice fed with NCD or HFD both showed insulin resistance after PS exposure accompanied by increased plasma lipopolysaccharide and pro-inflammatory cytokines, such as tumor necrosis factor-α and interleukin-1β. Exposure to PS also resulted in a significant decrease in the richness and diversity of gut microbiota, particularly an increase in the relative abundance of Gram-negative bacteria, such as Prevotellaceae and Enterobacteriaceae. The authors also demonstrated the presence of inhibition of the insulin signaling pathway in the liver of PS exposed mice. Taken together, these data suggest that PS might be a potential environmental contaminant that causes metabolic diseases associated with insulin resistance. Micro-nanoplastics absorbed by the intestine could also have important effects on lipid digestion and absorption [78].

The morphological alterations found in the placental cells could be the result of a prolonged attempt to remove and destroy the plastic particles inside the placental tissue. It is probable that the cells monotonously process threats from foreign bodies, of any organic or inorganic nature, but being faced with a practically indestructible material such as plastic, continuous prolonged enzymatic decommissioning leads to an accumulation of toxic substances, which determines a persistent state of cellular alarm CDR (cell danger response). The CDR is the evolutionarily conserved metabolic response that protects cells and hosts from harm. It is triggered by encounters with chemical, physical, or biological threats that exceed the cellular capacity for homeostasis. The resulting metabolic mismatch between available resources and functional capacity produces a cascade of changes in cellular electron flow, oxygen consumption, redox, membrane fluidity, lipid dynamics, bioenergetics, carbon and sulfur resource allocation, protein folding and aggregation, vitamin availability, metal homeostasis, indole, pterin, 1-carbon and polyamine metabolism, and polymer formation [79], which causes fetal phenotypic alterations in the long-term (Figure 8).

Based on the information gathered above, we note that the presence of particles compatible with MPs observed for the first time by VP-SEM and TEM, in at term human placenta samples, could contribute to the activation of pathological traits, such as oxidative stress, apoptosis, and inflammation, characteristic of metabolic disorders which underlie future diseases, such as obesity, diabetes, metabolic syndrome, and many other pathologies, which have their roots in oxidative stress damage and organelle dysfunction.

## 4. Conclusions

In this study, we demonstrated, for the first time, by VP-SEM and TEM, the presence and localization in the intra/extracellular compartment of fragments compatible with MPs in the human placenta and we hypothesized a possible correlation between their presence and the important ultrastructural alterations of intracytoplasmic organelles. We found changes of the morphology of endoplasmic reticulum and mitochondria that had never been reported in normal healthy at term pregnancies until today. These alterations are instead characteristic only of pathological states, according to numerous characterizing studies on human placenta conducted in the 1990s.

The modification of the structure of the organelles together with the presence of the MPs, in all the samples examined, is a very important item, since endoplasmic reticulum stress and mitochondrial dysfunction could play a decisive role in the progression of human non-transmissible disease. Hence, this study provides data support for further research on the potential role of MPs in different aspects of human disease.

This study might also contribute to the demonstration that the environmental pollution from plastic is partially responsible for the epidemic of non-transmissible disease that characterizes the modern world. The invisible epidemic of non-transmissible disease is an under-appreciated cause of poverty and hinders the economic development of many countries. Adding plastic to the preventable causes (through better, more prudent use and reduced production) of this epidemic, together with fasting blood glucose, blood pressure, hypertension, cholesterol, harmful use of alcohol, insufficient physical activity, overweight/obesity, and tobacco use, would greatly improve the living conditions on our planet.

Further investigation will aim to better describe the correlation between the presence of MPs and the structural changes we have found, and to verify endoplasmic reticulum and mitochondrial stress with their related markers, especially from the point of view of cellular function.

## Figures and Tables

**Figure 1 ijerph-19-11593-f001:**
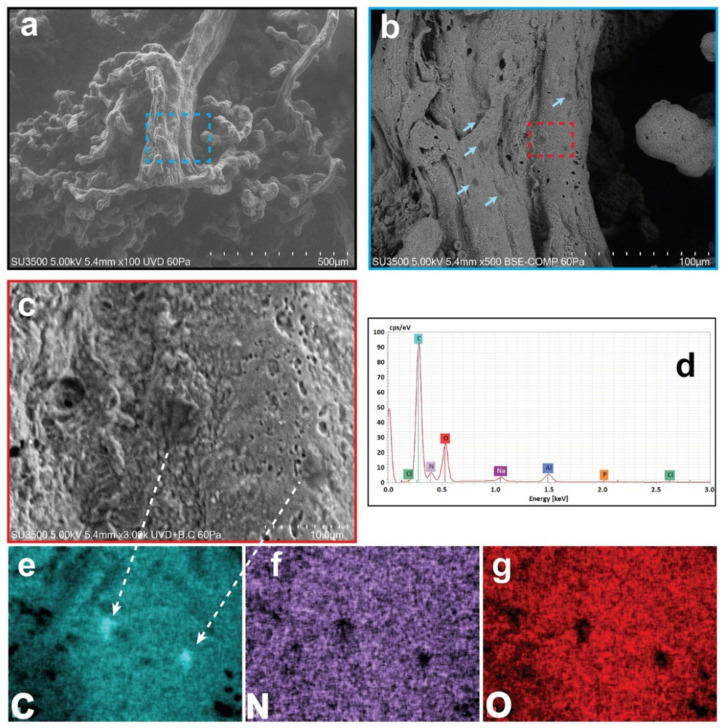
**Surface morpho-chemical study of MPs on stem villus in hydrated human placenta (VP-SEM-dEDS).** (**a**) Stem villus with intermediate and terminal villous ramifications. (**b**) MPs deposited on magnified stem villus (blue arrowheads). (**c**) Two MPs on magnified region (red dotted square) and (**d**) corresponding EDS spectral image with (**e**–**g**) multi-elemental mapping images below (C, carbon; N, nitrogen; O, oxygen).

**Figure 2 ijerph-19-11593-f002:**
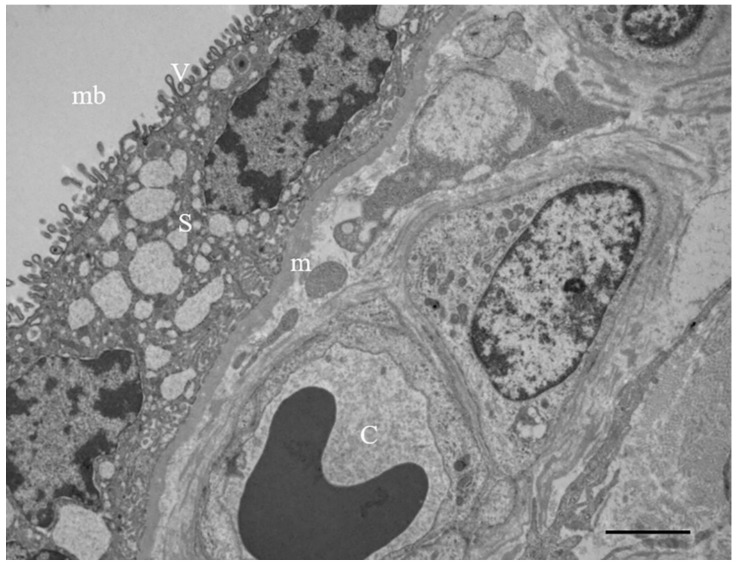
**Microphotography of an ultrathin sections of a terminal villus in at term placenta, by TEM.** Syncytiotrophoblast (S), covered by microvilli (v), is a continuous and uninterrupted multinuclear surface with the absence of cellular borders. It is separated from the stromal connective tissue by a basement membrane (m). fetal capillary (C); maternal blood (mb). Bar = 2 µm.

**Figure 3 ijerph-19-11593-f003:**
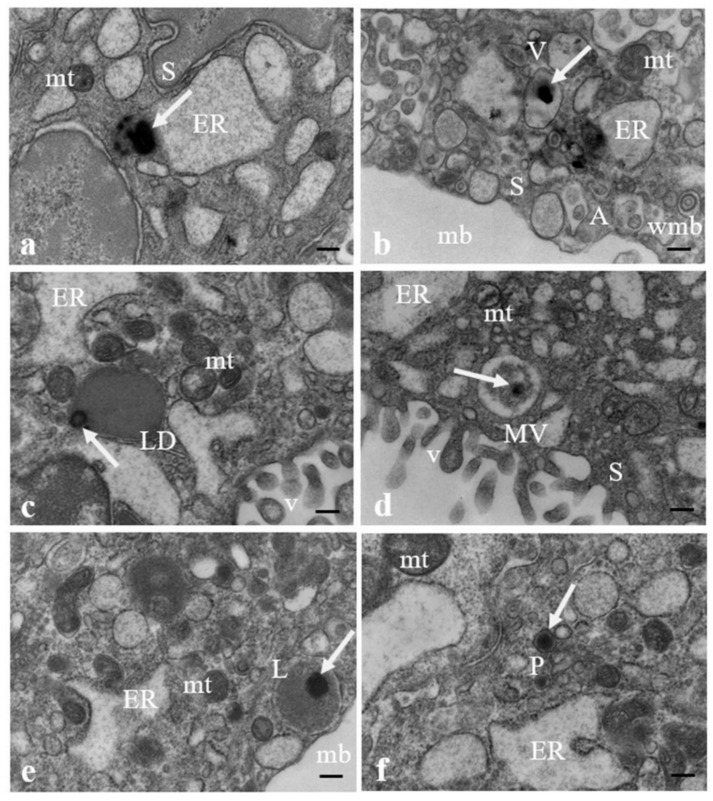
**MPs in human placenta**. (**a**) MPs (arrows) are found inside the syncytiotrophoblast (S), free in the cytoplasm and localized in a structure resembling (**b**) a vacuole (V), (**c**) a lipid droplet (LD), (**d**) a multivesicular body (MV), (**e**) a lysosome (L), (**f**) a peroxisome (P). In all images endoplasmic reticulum (ER) appears dilated. Swollen mitochondria (mt), whorled membranous bodies (wmb) and autolysosomes (A), particularly evident in (**b**), containing mitochondria, mitochondrial remnants and other structures also occur. maternal blood (mb); microvilli (v). Bar = 200 nm.

**Figure 4 ijerph-19-11593-f004:**
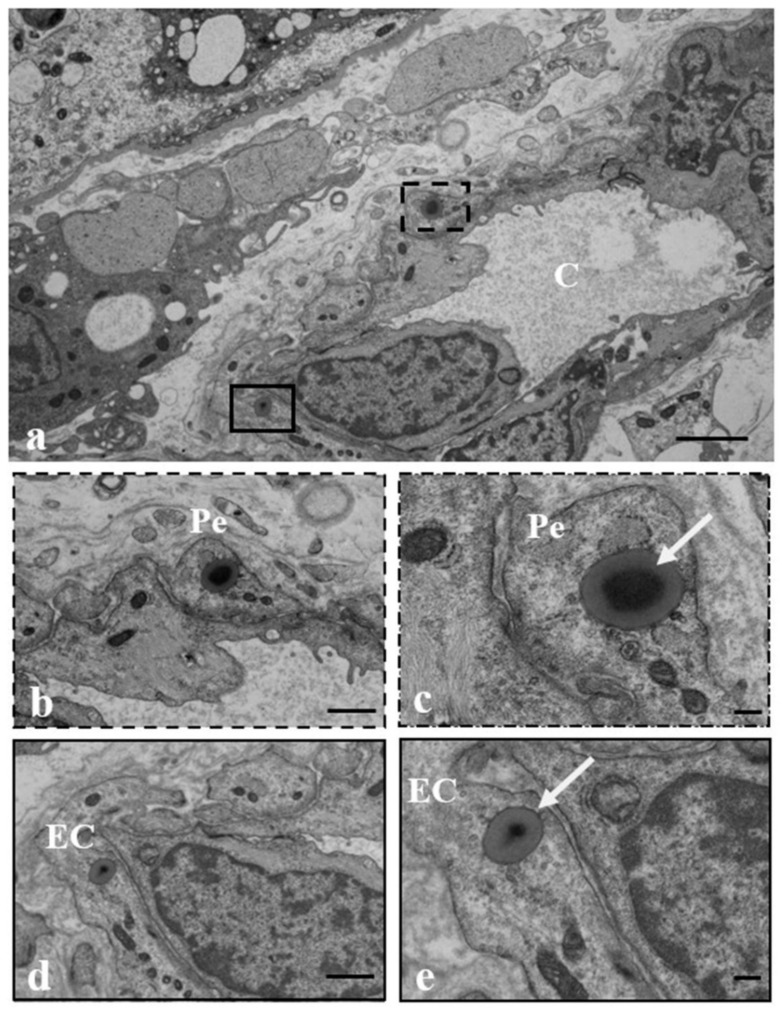
**MPs in the fetal microvessels of human placenta.** (**a**) Micrographs of a fetal microvessel (C) in which MPs (arrows) are (**b**,**c**) encapsuled in the cytoplasm of both the pericyte (Pe) that is lining the outer surface of endothelial cells (ECs) surrounding the sinusoid and (**d**,**e**) the ECs. Bar = 2 µm (**a**); Bar = 1µm (**b**,**d**); Bar = 200 nm (**c**,**e**).

**Figure 5 ijerph-19-11593-f005:**
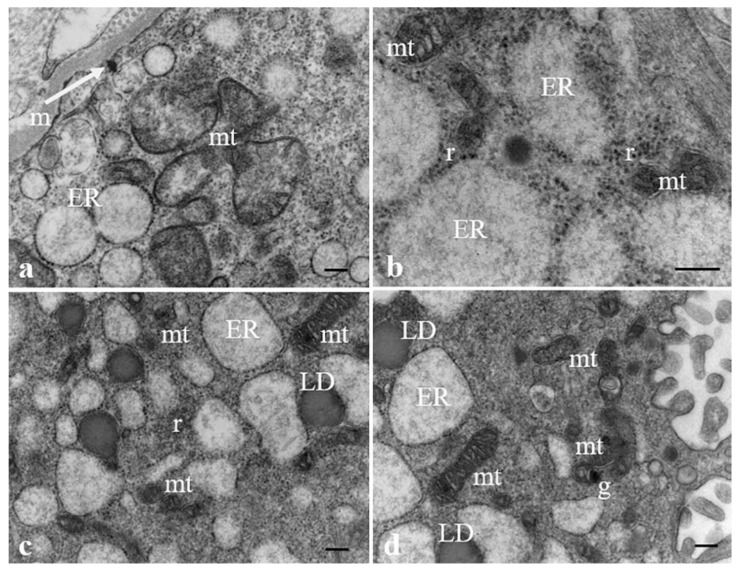
**Ultrastructural changes of organelles in the syncytiotrophoblast****layer.** (**a**) A MP (arrow) is present close to basement membrane. The endoplasmic reticulum (ER) appears dilated, with sparse ribosomes on the outer surface. (**b**) Ribosomes (r) result free in the cytoplasm and (**a**,**b**) swollen mitochondria (mt) next to (**c**,**d**) smaller, pycnotic and electrodense mitochondria are evident in the cytoplasm. (**d**) Mitochondrial matrix contains prominent electron-dense granules (g). Lipid droplets (LD) close to ER dilatated vesicles are also present (**c**,**d**). Bar, 200 nm.

**Figure 6 ijerph-19-11593-f006:**
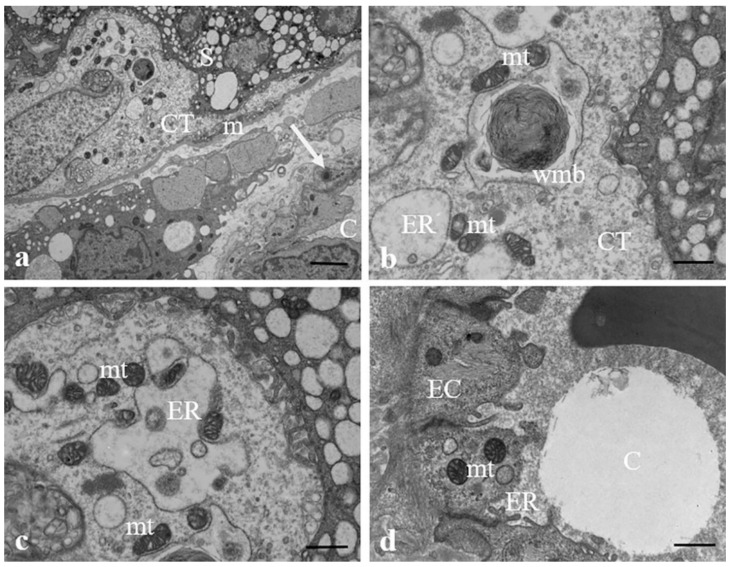
**Ultrastructural changes of organelles in the cytotrophoblast cells.** (**a**–**c**) Micrographs of the cytotrophoblast layer (CT) in which it is possible to observe in the cytoplasm of the cell dilated vesicles of ER, numerous electrodense swollen mitochondria (mt) and whorled membranous bodies (wmb) derived from involuting ER and other structures. (**d**) Changes in the structure of ER and mitochondria occur also in endothelial cells (ECs) surrounding the fetal capillaries (C). MP (arrow) in a pericyte is seen (**a**). Bar, 2 µm (**a**) and 800 nm (**b**–**d**).

**Figure 7 ijerph-19-11593-f007:**
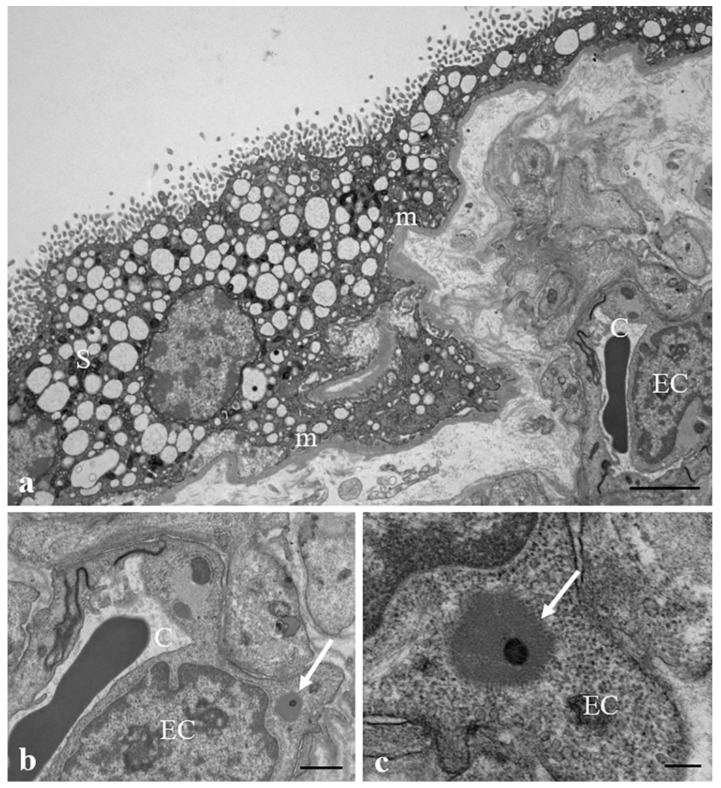
**MPs and narrowing of the fetal capillaries.** (**a**,**b**) Transmission electron micrographs of a narrowing of the fetal micro-vessel (C) in association with (**b**,**c**) the presence of a MP (arrow) inside the cytoplasm of an endothelial cell (EC). Bar, 2 µm (**a**), 800 nm (**b**), 200 nm (**c**).

**Figure 8 ijerph-19-11593-f008:**
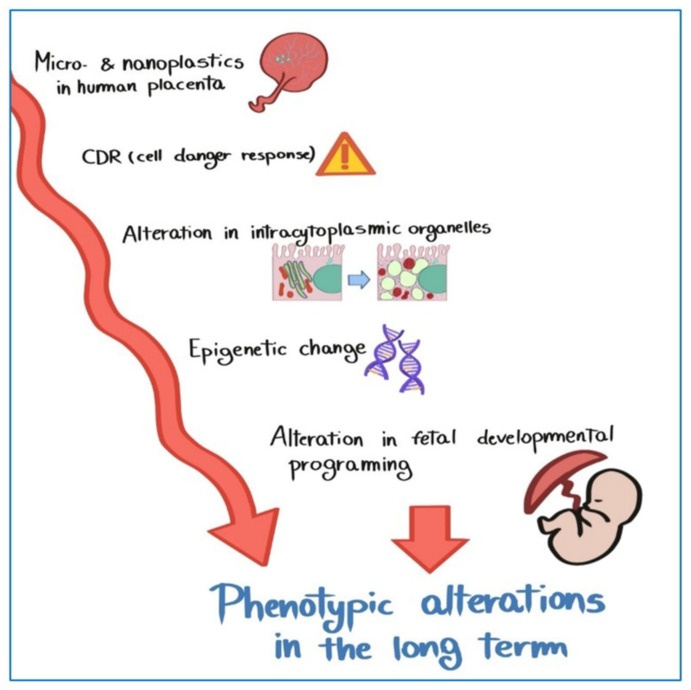
Hypotetical mechanism by which MPs could lead epigenetic changes that determine fetal phenotypic alterations in the long-term.

**Table 1 ijerph-19-11593-t001:** Age, type of childbirth, lifestyle, routine habits of plastic consumption and pathophysiological conditions of the enrolled patients.

Patient	1	2	3	4	5	6	7	8	9	10
**Age**	33	25	38	36	28	36	26	42	31	39
**Types of childbirth**	Natural	C-section	Natural	Natural	C-section	C-section	Natural	C-section	Natural	C-section
**Vegetarian**	No	No	No	No	No	No	No	No	No	No
**Seafood in the last 5 days**	No	No	No	No	No	No	No	No	No	Yes
**Pathologies**	Gestional Diabetes with insulin	No	No	No	No	No	No	Hypotiroidism	No	No
**Food in plastic wrapping**	Yes	Yes	Yes	No	Yes	Yes	Yes	Yes	Yes	Yes
**Drinks in plastic containers**	Yes	Yes	Yes	Yes	Yes	Yes	Yes	Yes	Yes	Yes
**Cosmetics with synthetic polymers**	Rilastil, Eucerin	Yes	Scrub Yves Roche	Scub Yves Roche	No	Scrub shower gel	No	No	Bionike	Veralab, Mustela, Dove
**Toothpastes with syntethic polymers**	Yes	Yes	Yes	Yes	Yes	Yes	Yes	Yes	Yes	Yes
**Chewing-gum**	No	No	Yes, once a month	No	Yes, once a month	No	Yes, once a month	No	No	No
**Smoke**	Ex	10cg/die	No	No	No	No	No	No	No	No
**Alcohol**	No	No	No	No	No	No	No	No	No	No

**Table 2 ijerph-19-11593-t002:** Presence of MPs in ultrathin sections of ten samples of human placenta.

Patient	1	2	3	4	5	6	7	8	9	10
**MPs**	++	+	++	++	+	+++	+++	++	++	++
**Villus layer localization**	outer/basement membrane	outer	outer	outer	outer	outer/inner	outer/inner	outer/basement membrane	outer	outer/inner
**Intracellular localization**	lysosomes/peroxisomes/lipid droplets	lysosomes	vacuoles	lysosomes/peroxisomes	lysosomes	multivesicular bodies/peroxisomes	multivesicular bodies/lipid droplets	multivesicular bodies	lysosomes/peroxisomes	lysosomes
**Extracellular localization**	stroma	stroma	stroma	stroma	stroma	stroma/sinusoids (pericytes)	stroma/sinusoids (pericytes/endothelial cells)	stroma	stroma	stroma/sinusoids (pericytes)
**ER dilatation**	++	+	++	++	+	+++	+++	+++	++	++
**Aggresomes**	+++	-	-	-	-	-	+	+	-	-
**Mitochondria damaged**	++ (swollen)	+ (swollen)	++ (swollen)	++ (swollen)	+ (swollen)	++ (swollen/ picnotic)	+++ (swollen/ picnotic)	++ (swollen/ picnotic)	++ (swollen)	++ (swollen/picnotic)
**Mitochondrial granules**	-	-	++	++	-	++	+++	++	++	+
**Whorled membranous bodies**	++ involution of endoplasmic reticulum/mitochondria	+ involution of mitochondria/microvilli	+++ involution of endoplasmic reticulum/mitochondria/microvilli	++ involution of endoplasmic reticulum/mitochondria	+ involution of mitochondria/microvilli	++ involution of endoplasmic reticulum/mitochondria/microvilli	+++ involution of endoplasmic reticulum/mitochondria	++ involution of endoplasmic reticulum/mitochondria/microvilli	++ involution of endoplasmic reticulum/mitochondria	++ involution of endoplasmic reticulum/mitochondria

## Data Availability

Not applicable.

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
