# Peer review of "Deeply in Plasticenta: Presence of Microplastics in the Intracellular Compartment of Human Placentas"

_ijerph, 2022, doi:10.3390/ijerph191811593_

Round 1
Reviewer 2 Report
Reviewer 1:
I recommend a major amendment at this level.
General comments:
I reviewed the manuscript entitled “Deeply in Plasticenta: Presence of microplastics in the intracellular compartment of human placentas”. The work carried out in the manuscript is interesting and focuses on locating MPs within the cellular compartment in the placenta and understanding whether their presence and location are associated with structural changes in cell organelles. Overall the manuscript is well written and easy to follow. There are adequate results presented in this manuscript but no justification or explanations for why results are so a mere reference to authors does not do justice. Authors must explain or make assumptions as to why the results are so etc. Also, a clear comparison with recent previous work should be stated in a more scientific manner. Then, citation lumps need to be resolved. Discussion of the results should provide useful insights. The conclusion part is needed more improvement in terms of writing and explaining the significance of the study. Highlights are necessary for this journal. It is better to do not to use the first person's pronoun. Do not use "we, us, or our" throughout the paper. It is recommended that the authors work with a science editor who is proficient in the Native English language to improve the organization and delivery of some portions of the manuscript. This will help improve the readability and help articulate better the relevance of the authors' work. Specific comments are given below:
General Comments:
Title: Ok.
Abstract:
Abstract not written well. What is the unique contribution of this paper? The indication of this feature should start already from the abstract. Otherwise, not too many readers will bother reading the paper after looking at the abstract. Please add an indication of the achievements from your study that are relevant to the journal scope. Please be concise - maximum 1-2 lines. Please incorporate to give meaning to the work.
Introduction:
Too short. Please extend at least one and a half pages. The novelty and importance of this study are not clearly described in the introduction. The introduction is not at a satisfactory level, only the authors reported several studies. It should follow the state of the art in this field and review what has been done, for supporting the research gap and the significance of this study. Please improve the state-of-the-art overview, to clearly show the progress beyond the state of the art. The lack of proper justification creates the wrong impression that the authors are unaware of the recent developments. The literature review should clarify the "contribution" of your study. The authors failed to present the study debates and failed to discuss the debates. In general, the authors should present the specific debate for your study. Please provide a table and compare your results with others. The relevant reference may be of interest to the author according to below:
https://www.sciencedirect.com/science/article/abs/pii/S0025326X18303291
Please eliminate the use of redundant words. Eg. In this way, Recently, Respectively, therefore, currently, thus, hence, finally, to do this, first, in order, however, moreover, nowadays, today, consequently, in addition, additionally, furthermore. Please revise all similar cases, as removing these term(s) would not significantly affect the meaning of the sentence. This will keep the manuscript as CONCISE as possible. Please check ALL. Avoid beginning or ending a sentence with one or a few words, they are usually redundant. Kindly revise all.
Materials and Methods:
Please avoid having one heading after another with no discussion in between as in the case of Sections 2 and 2.1. Kindly inspect the entire document for similar instances and revise accordingly. Please add in the beginning your scientific hypothesis. In the course of describing the performed actions, please provide reader guidance, sufficient for understanding why those actions have been performed. Any references in this section???
Results and discussion:
The structure of this work should be reorganized. For example, a Section of results should be combined with the Discussion. The authors are suggested to have the results and discussion part together. The authors need to present how the results can be validated and verified. Please cite and use recent references and compared them with previous literature. The discussion and interpretation of results do not clearly explain their impact on the literature and the field. Please improve the graphics quality. Please add at least one data in a table into the results section.
Conclusions:
The conclusions drawn in this manuscript are quite basic. In my opinion, be written in a more precise manner and not present the results only. The conclusion section appears to be just a detailed summary of results/observations. All conclusions must be convincing statements on what was found to be novel, and impactful based on the strong support of the data/results/discussion. Authors should conclude by stating the highlights of the study, benefits, and the applicability of your findings/results for further work and recommendations.
References:
Please check the reference section carefully and correct the inconsistency.
Reviewer 3 Report
I have read the manuscript entitled: Deeply in Plasticenta: Presence of microplastics in the intracellular compartment of human placentas. Its a very important judging from the excessive pollution that humans have created, although this is affecting our environmental quality and health. I find the manuscript to have been properly written and it has produced very important results.
However, in a few places I found a few minor problems that can be resolved relatively eazy. The problems are explained as follows.
Observation No 1. The authors have written as follows (Line 97-98):
'For the placentas collection, we used a plastic free-kit, and the sampling has been conducted following predefined instruction to avoid any contamination with plastic or synthetic fibers'.
My Comment: Although the sentence appear well written, can you please explain a little bit more on what do you mean by a 'pre-defined instruction'? In other word, I am asking you to explain further but not too much detail.
Observation No .2: In line 311, the authors have written as follows:
'Thanks to our instrumentation and the existing literature on MPs chemical spectrum.........."
My Comment: Can you please specify the specific literature that you are referring to? Please include the in-text referencing of such literature so that I am able to follow on this line of discussion. The journal uses numbers in chronological order to account for reference consulted. l
Observation No 3 (Line 333): There is some inconsistency in how you have written the reference at the end of the following sentence: 'We hypothesized that MPs may be uptaken by the tissue and encapsulated inside the organelles after reaching the placenta through the gastrointestinal and/or respiratory tracts and skin, as we previously suggested by Ragusa et al., (2021) [23].
Reviewer 4 Report
The authors have identified through electron microscopy particles inside human placenta that are compatible with microplastics. The microplastic particles have been found in cytoplasm, lipid membranes, lipid droplets, multivesicular bodies, lysosomes and peroxisomes. The authors report also morphological alterations in endoplasmic reticulum and mitochondrial swelling.
The manuscript displays sound and relevant findings, especially considering that the findings have been reported in human samples. I have no major objections against publication of the manuscript. However, I think that the authors could consider a few minor topics before.
I understand that we are dealing with a histological manuscript and quantification is hard. However, I wonder how representative of the general situation the showed micrographs are.
How many slices per placenta were scanned?
Are there significant differences among the ten assessed placentas?
Would be possible to establish a kind of semi-quantification establishing an approximated number of microplastic particles per slide?
Are there differences between vaginal and cesarean delivering?
It is known that there are differences between placentas carrying a male baby and placentas carrying a female baby. Are there any differences in these cases?
I know that 10 is a relatively small size but, is it possible to establish differences in plastic exposure of mothers based on the questionnaires? Is yes, can these differences be correlated with semi quantitative differences in number of detected microplastic particles.
Round 2
Reviewer 2 Report
Reviewer 2:
I have reviewed the revised version manuscript entitled” Deeply in Plasticenta: Presence of microplastics in the intracellular compartment of human placentas”. The work is interesting and it falls within the scope of the journal. Moreover, the authors adequately answered the queries of the reviewer. The paper has been improved and can be accepted. I do not have further comments.